# Probabilistic Model-Agnostic Meta-Learning

**Chelsea Finn**[*], **Kelvin Xu**[*], **Sergey Levine**
UC Berkeley
{cbfinn,kelvinxu,svlevine}@eecs.berkeley.edu

## Abstract

Meta-learning for few-shot learning entails acquiring a prior over previous tasks and experiences, such that new tasks be learned from small amounts of data. However, a critical challenge in few-shot learning is task ambiguity: even when a powerful prior can be meta-learned from a large number of prior tasks, a small dataset for a new task can simply be too ambiguous to acquire a single model (e.g., a classifier) for that task that is accurate. In this paper, we propose a probabilistic meta-learning algorithm that can sample models for a new task from a model distribution. Our approach extends model-agnostic meta-learning, which adapts to new tasks via gradient descent, to incorporate a parameter distribution that is trained via a variational lower bound. At meta-test time, our algorithm adapts via a simple procedure that injects noise into gradient descent, and at meta-training time, the model is trained such that this stochastic adaptation procedure produces samples from the approximate model posterior. Our experimental results show that our method can sample plausible classifiers and regressors in ambiguous few-shot learning problems. We also show how reasoning about ambiguity can also be used for downstream active learning problems.

## 1 Introduction

Learning from a few examples is a key aspect of human intelligence. One way to make it possible to acquire solutions to complex tasks from only a few examples is to leverage past experience to learn a prior over tasks. The process of learning this prior entails discovering the shared structure across different tasks from the same family, such as commonly occurring visual features or semantic cues. Structure is useful insofar as it yields efficient learning of new tasks – a mechanism known as learning-to-learn, or meta-learning [3]. However, when the end goal of few-shot meta-learning is to learn solutions to new tasks from small amounts of data, a critical issue that must be dealt with is task *ambiguity*: even with the best possible prior, there might simply not be enough information in the examples for a new task to resolve that task with high certainty. It is therefore quite desireable to develop few-shot meta-learning methods that can propose multiple potential solutions to an ambiguous few-shot learning problem. Such a method could be used to evaluate uncertainty (by measuring agreement between the samples), perform active learning, or elicit direct human supervision about which sample is preferable. For example, in safety-critical applications, such as few-shot medical image classification, uncertainty is crucial for determining if the learned classifier should be trusted. When learning from such small amounts of data, uncertainty estimation can also help predict if additional data would be beneficial for learning and improving the estimate of the rewards. Finally, while we do not experiment with this in this paper, we expect that modeling this ambiguity will be helpful for reinforcement learning problems, where it can be used to aid in exploration.

While recognizing and accounting for ambiguity is an important aspect of the few-shot learning problem, it is challenging to model when scaling to high-dimensional data, large function approximators, and multimodal task structure. Representing distributions over functions is relatively straightforward

---

[*]First two authors contributed equally.

when using simple function approximators, such as linear functions, and has been done extensively in early few-shot learning approaches using Bayesian models [39, 7]. But this problem becomes substantially more challenging when reasoning over high-dimensional function approximators such as deep neural networks, since explicitly representing expressive distributions over thousands or millions of parameters if often intractable. As a result, recent more scalable approaches to few-shot learning have focused on acquiring deterministic learning algorithms that disregard ambiguity over the underlying function. Can we develop an approach that has the benefits of both classes of few-shot learning methods – scalability and uncertainty awareness? To do so, we build upon tools in amortized variational inference for developing a probabilistic meta-learning approach.

In particular, our method builds on model-agnostic meta-learning (MAML) [9], a few shot meta-learning algorithm that uses gradient descent to adapt the model at meta-test time to a new few-shot task, and trains the model parameters at meta-training time to enable rapid adaptation, essentially optimizing for a neural network initialization that is well-suited for few shot learning. MAML can be shown to retain the generality of black-box meta-learners such as RNNs [8], while being applicable to standard neural network architectures. Our approach extends MAML to model a distribution over prior model parameters, which leads to an appealing simple stochastic adaptation procedure that simply injects noise into gradient descent at meta-test time. The meta-training procedure then optimizes for this simple inference process to produce samples from an approximate model posterior.

The primary contribution of this paper is a reframing of MAML as a graphical model inference problem, where variational inference can provide us with a principled and natural mechanism for modeling uncertainty. Our approach enables sampling multiple potential solutions to a few-shot learning problem at meta-test time, and our experiments show that this ability can be used to sample multiple possible regressors for an ambiguous regression problem, as well as multiple possible classifiers for ambiguous few-shot attribute classification tasks. We further show how this capability to represent uncertainty can be used to inform data acquisition in a few-shot active learning problem.

## 2   Related Work

Hierarchical Bayesian models are a long-standing approach for few-shot learning that naturally allow for the ability to reason about uncertainty over functions [39, 7, 25, 43, 12, 4, 41]. While these approaches have been demonstrated on simple few-shot image classification datasets [24], they have yet to scale to the more complex problems, such as the experiments in this paper. A number of works have approached the problem of few-shot learning from a meta-learning perspective [35, 19], including black-box [33, 5, 42] and optimization-based approaches [31, 9]. While these approaches scale to large-scale image datasets [40] and visual reinforcement learning problems [28], they typically lack the ability to reason about uncertainty.

Our work is most related to methods that combine deep networks and probabilistic methods for few-shot learning [6, 15, 23]. One approach that considers hierarchical Bayesian models for few-shot learning is the neural statistician [6], which uses an explicit task variable to model task distributions. Our method is fully model agnostic, and directly samples model weights for each task for any network architecture. Our experiments show that our approach improves on MAML [9], which outperforms the model by Edwards and Storkey [6]. Other work that considers model uncertainty in the few-shot learning setting is the LLAMA method [15], which also builds on the MAML algorithm. LLAMA makes use of a local Laplace approximation for modeling the task parameters (post-update parameters), which introduces the need to approximate a high dimensional covariance matrix. We instead propose a method that approximately infers the pre-update parameters, which we make tractable through a choice of approximate posterior parameterized by gradient operations.

*Bayesian neural networks* [27, 18, 29, 1] have been studied extensively as a way to incorporate uncertainty into deep networks. Although exact inference in Bayesian neural networks is impractical, approximations based on backpropagation and sampling [16, 32, 20, 2] have been effective in incorporating uncertainty into the weights of generic networks. Our approach differs from these methods in that we explicitly train a hierarchical Bayesian model over weights, where a posterior task-specific parameter distribution is inferred at meta-test time conditioned on a learned weight prior and a (few-shot) training set, while conventional Bayesian neural networks directly learn only the posterior weight distribution for a single task. Our method draws on amortized variational inference methods [22, 21, 36] to make this possible, but the key modification is that the model and inference networks share the same parameters. The resulting method corresponds structurally to a Bayesian version of model-agnostic meta-learning [9].

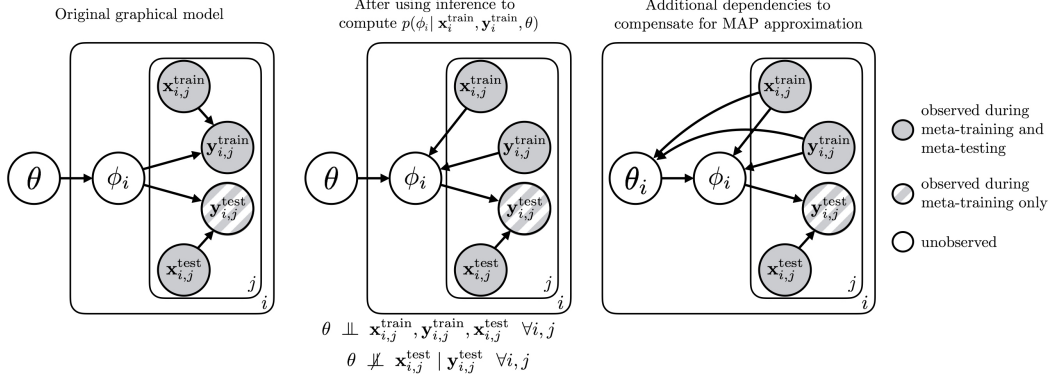

Figure 1: Graphical models corresponding to our approach. The original graphical model (left) is transformed into the center model after performing inference over $\phi_i$. We find it beneficial to introduce additional dependencies of the prior on the training data to compensate for using the MAP estimate to approximate $p(\phi_i)$, as shown on the right.

## 3 Preliminaries

In the meta-learning problem setting that we consider, the goal is to learn models that can learn new tasks from small amounts of data. To do so, meta-learning algorithms require a set of meta-training and meta-testing tasks drawn from some distribution $p(\mathcal{T})$. The key assumption of learning-to-learn is that the tasks in this distribution share common structure that can be exploited for faster learning of new tasks. Thus, the goal of the meta-learning process is to discover that structure. In this section, we will introduce notation and overview the model-agnostic meta-learning (MAML) algorithm [9].

Meta-learning algorithms proceed by sampling data from a given task, and splitting the sampled data into a set of a few datapoints, $\mathcal{D}^{\text{tr}}$ used for training the model and a set of datapoints for measuring whether or not training was effective, $\mathcal{D}^{\text{test}}$. This second dataset is used to measure few-shot generalization drive meta-training of the learning procedure. The MAML algorithm trains for few-shot generalization by optimizing for a set of initial parameters $\theta$ such that one or a few steps of gradient descent on $\mathcal{D}^{\text{tr}}$ achieves good performance on $\mathcal{D}^{\text{test}}$. Specifically, MAML performs the following optimization:

$$\min_{\theta} \sum_{\mathcal{T}_i \sim p(\mathcal{T})} \mathcal{L}(\theta - \alpha \nabla_\theta \mathcal{L}(\theta, \mathcal{D}^{\text{tr}}_{\mathcal{T}_i}), \mathcal{D}^{\text{test}}_{\mathcal{T}_i}) = \min_{\theta} \sum_{\mathcal{T}_i \sim p(\mathcal{T})} \mathcal{L}(\phi_i, \mathcal{D}^{\text{test}}_{\mathcal{T}_i})$$

where $\phi_i$ is used to denote the parameters updated by gradient descent and where the loss corresponds to negative log likelihood of the data. In particular, in the case of supervised classification with inputs $\{\mathbf{x}_j\}$, their corresponding labels $\{\mathbf{y}_j\}$, and a classifier $f_\theta$, we will denote the negative log likelihood of the data under the classifier as $\mathcal{L}(\theta, \mathcal{D}) = -\sum_{(\mathbf{x}_j, \mathbf{y}_j) \in \mathcal{D}} \log p(\mathbf{y}_j | \mathbf{x}_j, \theta)$. This corresponds to the cross entropy loss function.

## 4 Method

Our goal is to build a meta-learning method that can handle the uncertainty and ambiguity that occurs when learning from small amounts of data, while scaling to highly-expressive function approximators such as neural networks. To do so, we set up a graphical model for the few-shot learning problem. In particular, we want a hierarchical Bayesian model that includes random variables for the prior distribution over function parameters, $\theta$, the distribution over parameters for a particular task, $\phi_i$, and the task training and test datapoints. This graphical model is illustrated in Figure 1 (left), where tasks are indexed over $i$ and datapoints are indexed over $j$. We will use the shorthand $\mathbf{x}_i^{\text{tr}}, \mathbf{y}_i^{\text{tr}}, \mathbf{x}_i^{\text{test}}, \mathbf{y}_i^{\text{test}}$ to denote the sets of datapoints $\{\mathbf{x}_{i,j}^{\text{tr}} | \forall j\}, \{\mathbf{y}_{i,j}^{\text{tr}} | \forall j\}, \{\mathbf{x}_{i,j}^{\text{test}} | \forall j\}, \{\mathbf{y}_{i,j}^{\text{test}} | \forall j\}$ and $\mathcal{D}_i^{\text{tr}}, \mathcal{D}_i^{\text{test}}$ to denote $\{\mathbf{x}_i^{\text{tr}}, \mathbf{y}_i^{\text{tr}}\}$ and $\{\mathbf{x}_i^{\text{test}}, \mathbf{y}_i^{\text{test}}\}$.

### 4.1 Gradient-Based Meta-Learning with Variational Inference

In the graphical model in Figure 1, the predictions for each task are determined by the task-specific model parameters $\phi_i$. At meta-test time, these parameters are influenced by the prior $p(\phi_i | \theta)$, as well as by the observed training data $\mathbf{x}^{\text{tr}}, \mathbf{y}^{\text{tr}}$. The test inputs $\mathbf{x}^{\text{test}}$ are also observed, but the test outputs $\mathbf{y}^{\text{test}}$, which need to be predicted, are not observed. Note that $\phi_i$ is thus independent of $\mathbf{x}^{\text{test}}$, but not of

$\mathbf{x}^{\text{tr}}, \mathbf{y}^{\text{tr}}$. Therefore, posterior inference over $\phi_i$ must take into account both the evidence (training set) and the prior imposed by $p(\theta)$ and $p(\phi_i|\theta)$. Conventional MAML can be interpreted as approximating maximum a posteriori inference under a simplified model where $p(\theta)$ is a delta function, and inference is performed by running gradient descent on $\log p(\mathbf{y}^{\text{tr}}|\mathbf{x}^{\text{tr}}, \phi_i)$ for a fixed number of iterations starting from $\phi_i^0 = E[\theta]$ [15]. The corresponding distribution $p(\phi_i|\theta)$ is approximately Gaussian, with a mean that depends on the step size and number of gradient steps. When $p(\theta)$ is not deterministic, we must make a further approximation to account for the random variable $\theta$.

One way we can do this is by using structured variational inference. In structured variational inference, we approximate the distribution over the hidden variables $\theta$ and $\phi_i$ for each task with some approximate distribution $q_i(\theta, \phi_i)$. There are two reasonable choices we can make for $q_i(\theta, \phi_i)$. First, we can approximate it as a product of independent marginals, according to $q_i(\theta, \phi_i) = q_i(\theta)q_i(\phi_i)$. However, this approximation does not permit uncertainty to propagate effectively from $\theta$ to $\phi_i$. A more expressive approximation is the structured variational approximation $q_i(\theta, \phi_i) = q_i(\theta)q_i(\phi_i|\theta)$. We can further avoid storing a separate variational distribution $q_i(\phi_i|\theta)$ and $q_i(\theta)$ for each task $\mathcal{T}_i$ by employing an amortized variational inference technique [22, 21, 36], where we instead set $q_i(\phi_i|\theta) = q_\psi(\phi_i|\theta, \mathbf{x}_i^{\text{tr}}, \mathbf{y}_i^{\text{tr}}, \mathbf{x}_i^{\text{test}}, \mathbf{y}_i^{\text{test}})$, where $q_\psi$ is defined by some function approximator with parameters $\psi$ that takes $\mathbf{x}_i^{\text{tr}}, \mathbf{y}_i^{\text{tr}}$ as input, and the same $q_\psi$ is used for all tasks. Similarly, we can define $q_i(\theta)$ as $q_\psi(\theta|\mathbf{x}_i^{\text{tr}}, \mathbf{y}_i^{\text{tr}}, \mathbf{x}_i^{\text{test}}, \mathbf{y}_i^{\text{test}})$. We can now write down the variational lower bound on the log-likelihood as

$$\log p(\mathbf{y}_i^{\text{test}}|\mathbf{x}_i^{\text{test}}, \mathbf{x}_i^{\text{tr}}, \mathbf{y}_i^{\text{tr}}) \geq \underset{\theta, \phi_i \sim q_\psi}{\mathbb{E}}\Big[\log p(\mathbf{y}_i^{\text{tr}}|\mathbf{x}_i^{\text{tr}}, \phi_i) + \log p(\mathbf{y}_i^{\text{test}}|\mathbf{x}_i^{\text{test}}, \phi_i) + \log p(\phi_i|\theta) + \log p(\theta)\Big] +$$
$$\mathcal{H}(q_\psi(\phi_i|\theta, \mathbf{x}_i^{\text{tr}}, \mathbf{y}_i^{\text{tr}}, \mathbf{x}_i^{\text{test}}, \mathbf{y}_i^{\text{test}})) + \mathcal{H}(q_\psi(\theta|\mathbf{x}_i^{\text{tr}}, \mathbf{y}_i^{\text{tr}}, \mathbf{x}_i^{\text{test}}, \mathbf{y}_i^{\text{test}})).$$

The likelihood terms on the first line can be evaluated efficiently: given a sample $\theta, \phi_i \sim q(\theta, \phi_i|\mathbf{x}_i^{\text{tr}}, \mathbf{y}_i^{\text{tr}}, \mathbf{x}_i^{\text{test}}, \mathbf{y}_i^{\text{test}})$, the training and test likelihoods simply correspond to the loss of the network with parameters $\phi_i$. The prior $p(\theta)$ can be chosen to be Gaussian, with a learned mean and (diagonal) covariance to provide for flexibility to choose the prior parameters. This corresponds to a Bayesian version of the MAML algorithm. We will define these parameters as $\boldsymbol{\mu}_\theta$ and $\boldsymbol{\sigma}_\theta^2$. Lastly, $p(\phi_i|\theta)$ must be chosen. This choice is more delicate. One way to ensure a tractable likelihood is to use a Gaussian with mean $\theta$. This choice is reasonable, because it encourages $\phi_i$ to stay close to the prior parameters $\phi_i$, but we will see in the next section how a more expressive implicit conditional can be obtained using gradient descent, resulting in a procedure that more closely resembles the original MAML algorithm while still modeling the uncertainty. Lastly, we must choose a form for the inference networks $q_\psi(\phi_i|\theta, \mathbf{x}_i^{\text{tr}}, \mathbf{y}_i^{\text{tr}}, \mathbf{x}_i^{\text{test}}, \mathbf{y}_i^{\text{test}})$ and $q_\psi(\theta|\mathbf{x}_i^{\text{tr}}, \mathbf{y}_i^{\text{tr}}, \mathbf{x}_i^{\text{test}}, \mathbf{y}_i^{\text{test}})$. They must be chosen so that their entropies on the second line of the above equation are tractable. Furthermore, note that both of these distributions model very high-dimensional random variables: a deep neural network can have hundreds of thousands or millions of parameters. So while we can use an arbitrary function approximator, we would like to find a scalable solution.

One convenient solution is to allow $q_\psi$ to reuse the learned mean of the prior $\boldsymbol{\mu}_\theta$. We observe that adapting the parameters with gradient descent is a good way to update them to a given training set $\mathbf{x}_i^{\text{tr}}, \mathbf{y}_i^{\text{tr}}$ and test set $\mathbf{x}_i^{\text{test}}, \mathbf{y}_i^{\text{test}}$, a design decision similar to one made by Fortunato et al. [11]. We propose an inference network of the form

$$q_\psi(\theta|\mathbf{x}_i^{\text{tr}}, \mathbf{y}_i^{\text{tr}}, \mathbf{x}_i^{\text{test}}, \mathbf{y}_i^{\text{test}}) = \mathcal{N}(\boldsymbol{\mu}_\theta + \boldsymbol{\gamma}_q \nabla_{\boldsymbol{\mu}_\theta} \log p(\mathbf{y}_i^{\text{tr}}|\mathbf{x}_i^{\text{tr}}, \boldsymbol{\mu}_\theta) + \boldsymbol{\gamma}_q \nabla_{\boldsymbol{\mu}_\theta} \log p(\mathbf{y}_i^{\text{test}}|\mathbf{x}_i^{\text{test}}, \boldsymbol{\mu}_\theta); \mathbf{v}_q),$$

where $\mathbf{v}_q$ is a learned (diagonal) covariance, and the mean has an additional parameter beyond $\boldsymbol{\mu}_\theta$, which is a "learning rate" vector $\boldsymbol{\gamma}_q$ that is pointwise multiplied with the gradient. While this choice may at first seem arbitrary, there is a simple intuition: the inference network should produce a sample of $\theta$ that is close to the posterior $p(\theta|\mathbf{x}_i^{\text{tr}}, \mathbf{y}_i^{\text{tr}}, \mathbf{x}_i^{\text{test}}, \mathbf{y}_i^{\text{test}})$. A reasonable way to arrive at a value of $\theta$ close to this posterior is to adapt it to *both* the training set and test set.[2] Note that this is only done during meta-training. It remains to choose $q_\psi(\phi_i|\theta, \mathbf{x}_i^{\text{tr}}, \mathbf{y}_i^{\text{tr}}, \mathbf{x}_i^{\text{test}}, \mathbf{y}_i^{\text{test}})$, which can also be formulated as a conditional Gaussian with mean given by applying gradient descent.

Although this variational distribution is substantially more compact in terms of parameters than a separate neural network, it only provides estimates of the posterior during meta-training. At meta-test time, we must obtain the posterior $p(\phi_i|\mathbf{x}_i^{\text{tr}}, \mathbf{y}_i^{\text{tr}}, \mathbf{x}_i^{\text{test}})$, without access to $\mathbf{y}_i^{\text{test}}$. We can train a separate set of inference networks to perform this operation, potentially also using gradient descent within the inference network. However, these networks do not receive any gradient information during

**Algorithm 1** Meta-training, differences from MAML in red
---
**Require:** $p(\mathcal{T})$: distribution over tasks
1: initialize $\Theta := \{\boldsymbol{\mu}_\theta, \boldsymbol{\sigma}_\theta^2, \mathbf{v}_q, \boldsymbol{\gamma}_p, \boldsymbol{\gamma}_q\}$
2: **while** not done **do**
3:     Sample batch of tasks $\mathcal{T}_i \sim p(\mathcal{T})$
4:     **for all** $\mathcal{T}_i$ **do**
5:         $\mathcal{D}^{\text{tr}}, \mathcal{D}^{\text{test}} = \mathcal{T}_i$
6:         Evaluate $\nabla_{\boldsymbol{\mu}_\theta} \mathcal{L}(\boldsymbol{\mu}_\theta, \mathcal{D}^{\text{test}})$
7:         Sample $\theta \sim q = \mathcal{N}(\boldsymbol{\mu}_\theta - \boldsymbol{\gamma}_q \nabla_{\boldsymbol{\mu}_\theta} \mathcal{L}(\boldsymbol{\mu}_\theta, \mathcal{D}^{\text{test}}), \mathbf{v}_q)$
8:         Evaluate $\nabla_\theta \mathcal{L}(\theta, \mathcal{D}^{\text{tr}})$
9:         Compute adapted parameters with gradient descent:
          $\phi_i = \theta - \alpha \nabla_\theta \mathcal{L}(\theta, \mathcal{D}^{\text{tr}})$
10:    Let $p(\theta|\mathcal{D}^{\text{tr}}) = \mathcal{N}(\boldsymbol{\mu}_\theta - \boldsymbol{\gamma}_p \nabla_{\boldsymbol{\mu}_\theta} \mathcal{L}(\boldsymbol{\mu}_\theta, \mathcal{D}^{\text{tr}}), \boldsymbol{\sigma}_\theta^2))$
11:    Compute $\nabla_\Theta \big( \sum_{\mathcal{T}_i} \mathcal{L}(\phi_i, \mathcal{D}^{\text{test}})$
                  $+ D_{\text{KL}}(q(\theta|\mathcal{D}^{\text{test}}) \| p(\theta|\mathcal{D}^{\text{tr}})))$
12:    Update $\Theta$ using Adam
---

---
**Algorithm 2** Meta-testing
---
**Require:** training data $\mathcal{D}_{\mathcal{T}}^{\text{tr}}$ for new task $\mathcal{T}$
**Require:** learned $\Theta$
1: Sample $\theta$ from the prior $p(\theta|\mathcal{D}^{\text{tr}})$
2: Evaluate $\nabla_\theta \mathcal{L}(\theta, \mathcal{D}^{\text{tr}})$
3: Compute adapted parameters with gradient descent:
   $\phi_i = \theta - \alpha \nabla_\theta \mathcal{L}(\theta, \mathcal{D}^{\text{tr}})$
---

meta-training, and may not work well in practice. In the next section we propose an even simpler and more practical approach that uses only a single inference network during meta-training, and none during meta-testing.

## 4.2 Probabilistic Model-Agnostic Meta-Learning Approach with Hybrid Inference

To formulate a simpler variational meta-learning procedure, we recall the probabilistic interpretation of MAML: as discussed by Grant et al. [15], MAML can be interpreted as approximate inference for the posterior $p(\mathbf{y}_i^{\text{test}}|\mathbf{x}_i^{\text{tr}}, \mathbf{y}_i^{\text{tr}}, \mathbf{x}_i^{\text{test}})$ according to

$$p(\mathbf{y}_i^{\text{test}}|\mathbf{x}_i^{\text{tr}}, \mathbf{y}_i^{\text{tr}}, \mathbf{x}_i^{\text{test}}) = \int p(\mathbf{y}_i^{\text{test}}|\mathbf{x}_i^{\text{test}}, \phi_i) p(\phi_i|\mathbf{x}_i^{\text{tr}}, \mathbf{y}_i^{\text{tr}}, \theta) d\phi_i \approx p(\mathbf{y}_i^{\text{test}}|\mathbf{x}_i^{\text{test}}, \phi_i^\star), \qquad (1)$$

where we use the maximum a posteriori (MAP) value $\phi_i^\star$. It can be shown that, for likelihoods that are Gaussian in $\phi_i$, gradient descent for a fixed number of iterations using $\mathbf{x}_i^{\text{tr}}, \mathbf{y}_i^{\text{tr}}$ corresponds exactly to maximum a posteriori inference under a Gaussian prior $p(\phi_i|\theta)$ [34]. In the case of non-Gaussian likelihoods, the equivalence is only locally approximate, and the exact form of the prior $p(\phi_i|\theta)$ is intractable. However, in practice this implicit prior can actually be preferable to an explicit (and simple) Gaussian prior, since it incorporates the rich nonlinear structure of the neural network parameter manifold, and produces good performance in practice [9, 15]. We can interpret this MAP approximation as inferring an approximate posterior on $\phi_i$ of the form $p(\phi_i|\mathbf{x}_i^{\text{tr}}, \mathbf{y}_i^{\text{tr}}, \theta) \approx \delta(\phi_i = \phi_i^\star)$, where $\phi_i^\star$ is obtained via gradient descent on the training set $\mathbf{x}_i^{\text{tr}}, \mathbf{y}_i^{\text{tr}}$ starting from $\theta$. Incorporating this approximate inference procedure transforms the graphical model in Figure 1 (a) into the one in Figure 1 (b), where there is now a factor over $p(\phi_i|\mathbf{x}_i^{\text{tr}}, \mathbf{y}_i^{\text{tr}}, \theta)$. While this is a crude approximation to the likelihood, it provides us with an empirically effective and simple tool that greatly simplifies the variational inference procedure described in the previous section, in the case where we aim to model a distribution over the global parameters $p(\theta)$. After using gradient descent to estimate $p(\phi_i \mid \mathbf{x}_i^{\text{tr}}, \mathbf{y}_i^{\text{tr}}, \theta)$, the graphical model is transformed into the model shown in the center of Figure 1. Note that, in this new graphical model, the global parameters $\theta$ are independent of $\mathbf{x}^{\text{tr}}$ and $\mathbf{y}^{\text{tr}}$ and are independent of $\mathbf{x}^{\text{test}}$ when $\mathbf{y}^{\text{test}}$ is not observed. Thus, we can now write down a variational lower bound for the logarithm of the *approximate* likelihood, which is given by

$$\log p(\mathbf{y}_i^{\text{test}}|\mathbf{x}_i^{\text{test}}, \mathbf{x}_i^{\text{tr}}, \mathbf{y}_i^{\text{tr}}) \geq E_{\theta \sim q_\psi} \left[ \log p(\mathbf{y}_i^{\text{test}}|\mathbf{x}_i^{\text{test}}, \phi_i^\star) + \log p(\theta) \right] + \mathcal{H}(q_\psi(\theta|\mathbf{x}_i^{\text{test}}, \mathbf{y}_i^{\text{test}})).$$

In this bound, we essentially perform approximate inference via MAP on $\phi_i$ to obtain $p(\phi_i|\mathbf{x}_i^{\text{tr}}, \mathbf{y}_i^{\text{tr}}, \theta)$, and use the variational distribution for $\theta$ only. Note that $q_\psi(\theta|\mathbf{x}_i^{\text{test}}, \mathbf{y}_i^{\text{test}})$ is not conditioned on the training set $\mathbf{x}_i^{\text{tr}}, \mathbf{y}_i^{\text{tr}}$ since $\theta$ is independent of it in the transformed graphical model. Analogously to the previous section, the inference network is given by

$$q_\psi(\theta|\mathbf{x}_i^{\text{test}}, \mathbf{y}_i^{\text{test}}) = \mathcal{N}(\boldsymbol{\mu}_\theta + \boldsymbol{\gamma}_q \nabla \log p(\mathbf{y}_i^{\text{test}}|\mathbf{x}_i^{\text{test}}, \boldsymbol{\mu}_\theta); \mathbf{v}_q).$$

To evaluate the variational lower bound during training, we can use the following procedure: first, we evaluate the mean by starting from $\boldsymbol{\mu}_\theta$ and taking one (or more) gradient steps on $\log p(\mathbf{y}_i^{\text{test}}|\mathbf{x}_i^{\text{test}}, \theta_{\text{current}})$, where $\theta_{\text{current}}$ starts at $\boldsymbol{\mu}_\theta$. We then add noise with variance $\mathbf{v}_q$, which

is made differentiable via the reparameterization trick [22]. We then take additional gradient steps on the training likelihood $\log p(\mathbf{y}_i^{\text{tr}}|\mathbf{x}_i^{\text{tr}}, \theta_{\text{current}})$. This accounts for the MAP inference procedure on $\phi_i$. Training of $\boldsymbol{\mu}_\theta$, $\boldsymbol{\sigma}_\theta^2$, and $\mathbf{v}_q$ is performed by backpropagating gradients through this entire procedure with respect to the variational lower bound, which includes a term for the likelihood $\log p(\mathbf{y}_i^{\text{test}}|\mathbf{x}_i^{\text{test}}, \mathbf{x}^{\text{tr}}, \mathbf{y}^{\text{tr}}, \phi_i^\star)$ and the KL-divergence between the sample $\theta \sim q_\psi$ and the prior $p(\theta)$. This meta-training procedure is detailed in Algorithm 1.

At meta-test time, the inference procedure is much simpler. The test labels are not available, so we simply sample $\theta \sim p(\theta)$ and perform MAP inference on $\phi_i$ using the training set, which corresponds to gradient steps on $\log p(\mathbf{y}_i^{\text{tr}}|\mathbf{x}_i^{\text{tr}}, \theta_{\text{current}})$, where $\theta_{\text{current}}$ starts at the sampled $\theta$. This meta-testing procedure is detailed in Algorithm 2.

### 4.3 Adding Additional Dependencies

In the transformed graphical model, the training data $\mathbf{x}_i^{\text{tr}}, \mathbf{y}_i^{\text{tr}}$ and the prior $\theta$ are conditionally independent. However, since we have only a crude approximation to $p(\phi_i \mid \mathbf{x}_i^{\text{tr}}, \mathbf{y}_i^{\text{tr}}, \theta)$, this independence often doesn't actually hold. We can allow the model to compensate for this approximation by additionally conditioning the learned prior $p(\theta)$ on the training data. In this case, the learned "prior" has the form $p(\theta_i|\mathbf{x}_i^{\text{tr}}, \mathbf{y}_i^{\text{tr}})$, where $\theta_i$ is now task-specific, but with global parameters $\boldsymbol{\mu}_\theta$ and $\boldsymbol{\sigma}_\theta^2$. We thus obtain the modified graphical model in Figure 1 (c). Similarly to the inference network $q_\psi$, we parameterize the learned prior as follows:

$$p(\theta_i|\mathbf{x}_i^{\text{tr}}, \mathbf{y}_i^{\text{tr}}) = \mathcal{N}(\boldsymbol{\mu}_\theta + \boldsymbol{\gamma}_p \nabla \log p(\mathbf{y}_i^{\text{tr}}|\mathbf{x}_i^{\text{tr}}, \boldsymbol{\mu}_\theta); \boldsymbol{\sigma}_\theta^2).$$

With this new form for distribution over $\theta$, the variational training objective uses the likelihood term $\log p(\theta_i|\mathbf{x}_i^{\text{tr}}, \mathbf{y}_i^{\text{tr}})$ in place of $\log p(\theta)$, but otherwise is left unchanged. At test time, we sample from $\theta \sim p(\theta|\mathbf{x}_i^{\text{tr}}, \mathbf{y}_i^{\text{tr}})$ by first taking gradient steps on $\log p(\mathbf{y}_i^{\text{tr}}|\mathbf{x}_i^{\text{tr}}, \theta_{\text{current}})$, where $\theta_{\text{current}}$ is initialized at $\boldsymbol{\mu}_\theta$, and then adding noise with variance $\boldsymbol{\sigma}_\theta^2$. Then, we proceed as before, performing MAP inference on $\phi_i$ by taking additional gradient steps on $\log p(\mathbf{y}_i^{\text{tr}}|\mathbf{x}_i^{\text{tr}}, \theta_{\text{current}})$ initialized at the sample $\theta$. In our experiments, we find that this more expressive distribution often leads to better performance.

## 5 Experiments

The goal of our experimental evaluation is to answer the following questions: (1) can our approach enable sampling from the distribution over potential functions underlying the training data?, (2) does our approach improve upon the MAML algorithm when there is ambiguity over the class of functions?, and (3) can our approach scale to deep convolutional networks? We study two illustrative toy examples and a realistic ambiguous few-shot image classification problem. For the both experimental domains, we compare MAML to our probabilistic approach. We will refer to our version of MAML as a PLATIPUS (Probabilistic LATent model for Incorporating Priors and Uncertainty in few-Shot learning), due to its unusual combination of two approximate inference methods: amortized inference and MAP. Both PLATIPUS and MAML use the same neural network architecture and the same number of inner gradient steps. We additionally provide a comparison on the MiniImagenet benchmark and specify the hyperparameters in the supplementary appendix.

**Illustrative 5-shot regression.** In this 1D regression problem, different tasks correspond to different underlying functions. Half of the functions are sinusoids, and half are lines, such that the task distribution is clearly multimodal. The sinusoids have amplitude and phase uniformly sampled from the range $[0.1, 5]$ and $[0, \pi]$, and the lines have the slope and intercept sampled in the range $[-3, 3]$. The input domain is uniform on $[-5, 5]$, and Gaussian noise with a standard deviation of $0.3$ is added to the labels. We trained both MAML and PLATIPUS for 5-shot regression. In Figure 2, we show the qualitative performance of both methods, where the ground truth underlying function is shown in gray and the datapoints in $\mathcal{D}^{\text{tr}}$ are shown as purple triangles. We show the function $f_{\phi_i}$ learned by MAML in black. For PLATIPUS, we sample 10 sets of parameters from $p(\phi_i|\theta)$ and plot the resulting functions in different colors. In the top row, we can see that PLATIPUS allows the model to effectively reason over the set of functions underlying the provided datapoints, with increased variance in parts of the function where there is more uncertainty. Further, we see that PLATIPUS is able to capture the multimodal structure, as the curves are all linear or sinusoidal.

A particularly useful application of uncertainty estimates in few-shot learning is estimating when more data would be helpful. In particular, seeing a large variance in a particular part of the input space suggests that more data would be helpful for learning the function in that part of the input space. On the bottom of Figure 2, we show the results for a single task at meta-test time with increasing numbers of training datapoints. Even though the model was only trained on training set sizes of 5

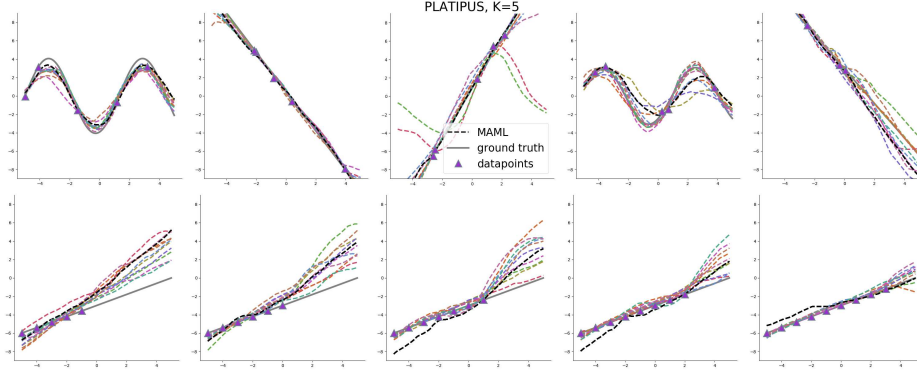

Figure 2: Samples from PLATIPUS trained for 5-shot regression, shown as colored dotted lines. The tasks consist of regressing to sinusoid and linear functions, shown in gray. MAML, shown in black, is a deterministic procedure and hence learns a single function, rather than reasoning about the distribution over potential functions. As seen on the bottom row, even though PLATIPUS is trained for 5-shot regression, it can effectively reason over its uncertainty when provided variable numbers of datapoints at test time (left vs. right).

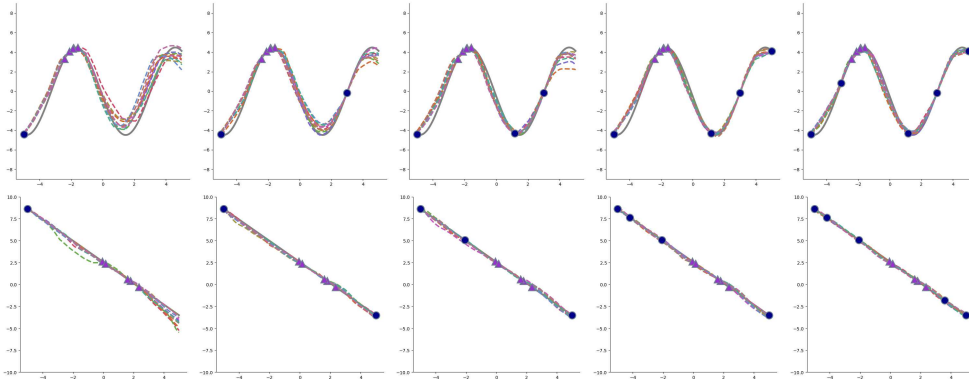

Figure 3: Qualitative examples from active learning experiment where the 5 provided datapoints are from a small region of the input space (shown as purple triangles), and the model actively asks for labels for new datapoints (shown as blue circles) by choosing datapoints with the largest variance across samples. The model is able to effectively choose points that leads to accurate predictions with only a few extra datapoints.

datapoints, we observe that PLATIPUS is able to effectively reduce its uncertainty as more and more datapoints are available. This suggests that the uncertainty provided by PLATIPUS can be used for approximately gauging when more data would be helpful for learning a new task.

**Active learning with regression.** To further evaluate the benefit of modeling ambiguity, we now consider an active learning experiment. In particular, the model can choose the datapoints that it wants labels for, with the goal of reaching good performance with a minimal number of additional datapoints. We performed this evaluation in the simple regression setting described previously. Models were given five initial datapoints within a constrained region of the input space. Then, each model selects up to 5 additional datapoints to be labeled. PLATIPUS chose each datapoint sequentially, choosing the point with maximal variance across the sampled regressors; MAML selected datapoints randomly, as it has no mechanism to model ambiguity. As seen in Figure 4, PLATIPUS is able to reduce its regression error to a much greater

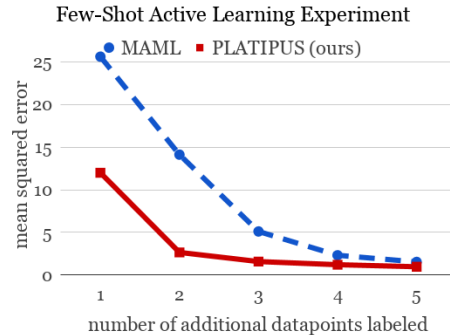

Figure 4: Active learning performance on regression after up to 5 selected datapoints. PLATIPUS can use it's uncertainty estimation to quickly decrease the error, while selecting datapoints randomly and using MAML leads to slower learning.

extent when given one to three additional queries, compared to MAML. We show qualitative results in Figure 3.

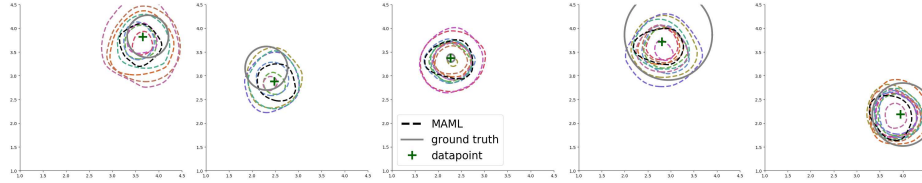

Figure 5: Samples from PLATIPUS for 1-shot classification, shown as colored dotted lines. The 2D classification tasks all involve circular decision boundaries of varying size and center, shown in gray. MAML, shown in black, is a deterministic procedure and hence learns a single function, rather than reasoning about the distribution over potential functions.

**Illustrative 1-Shot 2D classification.**   Next, we study a simple binary classification task, where there is a particularly large amount of ambiguity surrounding the underlying function: learning to learn from a single positive example. Here, the tasks consist of classifying datapoints in 2D within the range $[0, 5]$ with a circular decision boundary, where points inside the decision boundary are positive and points outside are negative. Different tasks correspond to different locations and radii of the decision boundary, sampled at uniformly at random from the ranges $[1.0, 4.0]$ and $[0.1, 2.0]$ respectively. Following Grant et al. [14], we train both MAML and PLATIPUS with $\mathcal{D}^{\text{tr}}$ consisting of a single positive example and $\mathcal{D}^{\text{test}}$ consisting of both positive and negative examples. We plot the results using the same scheme as before, except that we plot the decision boundary (rather than the regression function) and visualize the single positive datapoint with a green plus. As seen in Figure 5, we see that PLATIPUS captures a broad distribution over possible decision boundaries, all of which are roughly circular. MAML provides a single decision boundary of average size.

**Ambiguous image classification.**   The ambiguity illustrated in the previous settings is common in real world tasks where images can share multiple attributes. We study an ambiguous extension to the celebA attribute classification task. Our meta-training dataset is formed by sampling two attributes at random to form a positive class and taking the same number of random examples without either attribute to from the negative classes. To evaluate the ability to capture multiple decision boundaries while simultaneously obtaining good performance, we evaluate our method as follows: We sample from a test set of three attributes and a corresponding set of images with those attributes. Since the tasks involve classifying images that have two attributes, this task is ambiguous, and there are three possible combinations of two attributes that explain the training set. We sample models from our prior as described in Section 4 and assign each of the sampled models to one of the three possible tasks based on its log-likelihood. If each of the three possible tasks is assigned a nonzero number of samples, this means that the model effectively covers all three possible modes that explain the ambiguous training set. We can measure coverage and accuracy from this protocol. The coverage score indicates the average number of tasks (between 1 and 3) that receive at least one sample for each ambiguous training set, and the accuracy score is the average number of correct classifications on these tasks (according to the sampled models assigned to them). A highly random method will achieve good coverage but poor accuracy, while a deterministic method will have a coverage of 1. We additionally compute the log-likelihood across the ambiguous tasks which compares each method's ability to model all of the "modes". As is standard in amortized variational inference (e.g., with VAEs), we put a multiplier $\beta$ in front of the KL-divergence against the prior [17] in Algorithm 1. We find that larger values result in more diverse samples, at a modest cost in performance, and therefore report two different values of $\beta$ to illustrate this tradeoff.

Our results are summarized in Table 5 and Fig. 6. Our method attains better log-likelihood, and a comparable accuracy compared to standard MAML. More importantly, deterministic MAML only ever captures one mode for each ambiguous task, where the maximum is three. Our method on average captures closer to two modes on average. The qualitative analysis in Figure 6 illustrates[3] an example ambiguous training set, example images for the three possible two-attribute pairs that can correspond to this training set, and the classifications made by different sampled classifiers trained on the ambiguous training set. Note that the different samples each pay attention to different attributes, indicating that PLATIPUS is effective at capturing the different modes of the task.

## 6 Discussion and Future Work

We introduced an algorithm for few-shot meta-learning that enables simple and effective sampling of models for new tasks at meta-test time. Our algorithm, PLATIPUS, adapts to new tasks by running

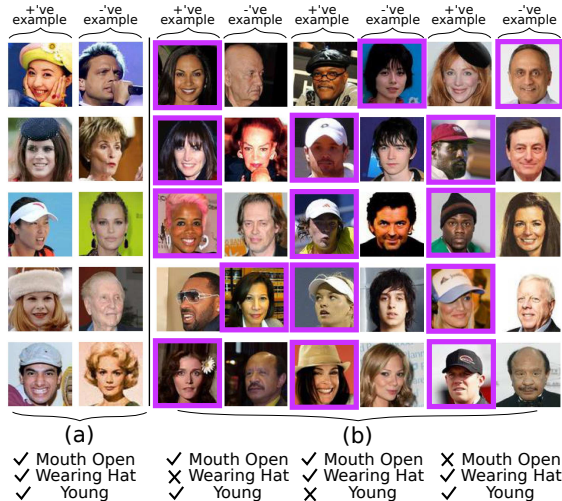

| +'ve example | -'ve example | +'ve example | -'ve example | +'ve example | -'ve example | +'ve example | -'ve example |

(a) (b)

✓ Mouth Open ✓ Wearing Hat ✓ Young | ✓ Mouth Open ✗ Wearing Hat ✓ Young | ✓ Mouth Open ✓ Wearing Hat ✗ Young | ✗ Mouth Open ✓ Wearing Hat ✓ Young

Figure 6: Sampled classifiers for an ambiguous meta-test task. In the meta-test training set (a), PLATIPUS observes five positives that share three attributes, and five negatives. A classifier that uses *any* two attributes can correctly classify the training set. On the right (b), we show the three possible two-attribute tasks that the training set can correspond to, and illustrate the labels (positive indicated by purple border) predicted by the best sampled classifier for that task. We see that different samples can effectively capture the three possible explanations, with some samples paying attention to hats (2nd and 3rd column) and others not (1st column).

| Ambiguous celebA (5-shot) | | | |
|---|---|---|---|
| | Accuracy | Coverage (max=3) | Average NLL |
| MAML | **89.00 ± 1.78**% | 1.00 ± 0.0 | 0.73 ± 0.06 |
| MAML + noise | 84.3 ± 1.60 % | 1.89 ± 0.04 | 0.68 ± 0.05 |
| **PLATIPUS (ours)** (KL weight = 0.05) | **88.34 ± 1.06** % | 1.59 ± 0.03 | 0.67 ± 0.05 |
| **PLATIPUS (ours)** (KL weight = 0.15) | 87.8 ± 1.03 % | **1.94 ± 0.04** | **0.56 ± 0.04** |

Table 1: Our method covers almost twice as many tasks compared to MAML, with comparable accuracy. MAML + noise is a method that adds noise to the gradient, but does not perform variational inference. This improves coverage, but results in lower accuracy average log likelihood. We bold results above the highest confidence interval lowerbound.

gradient descent with injected noise. During meta-training, the model parameters are optimized with respect to a variational lower bound on the likelihood for the meta-training tasks, so as to enable this simple adaptation procedure to produce approximate samples from the model posterior when conditioned on a few-shot training set. This approach has a number of benefits. The adaptation procedure is exceedingly simple, and the method can be applied to any standard model architecture. The algorithm introduces a modest number of additional parameters: besides the initial model weights, we must learn a variance on each parameter for the inference network and prior, and the number of parameters scales only linearly with the number of model weights. Our experimental results show that our method can be used to effectively sample diverse solutions to both regression and classification tasks at meta-test time, including with task families that have multi-modal task distributions. We additionally showed how our approach can be applied in settings where uncertainty can directly guide data acquisition, leading to better few-shot active learning.

Although our approach is simple and broadly applicable, it has potential limitations that could be addressed in future work. First, the current form of the method provides a relatively impoverished estimator of posterior variance, which might be less effective at gauging uncertainty in settings where different tasks have different degrees of ambiguity. In such settings, making the variance estimator dependent on the few-shot training set might produce better results, and investigating how to do this in a parameter efficient manner would be an interesting direction for future work. Another exciting direction for future research would be to study how our approach could be applied in RL settings for acquiring structured, uncertainty-guided exploration strategies in meta-RL problems.

### Acknowledgments

We thank Marvin Zhang and Dibya Ghosh for feedback on an early draft of this paper. This research was supported by an NSF Graduate Research Fellowship, NSF IIS-1651843, the Office of Naval Research, and NVIDIA.

## Footnotes

[2]In practice, we can use multiple gradient steps for the mean, but we omit this for notational simplicity.

[3]Additional qualitative results and code can be found at https://sites.google.com/view/probabilistic-maml/

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
