[Supplementary Material]

# Appendix

## A    Ambiguous CelebA Details

We construct our ambiguous few-shot variant of CelebA using the canonical splits to form the meta-train/val/test set. This gives us a split of $162770/19867/19962$ images respectively. We additionally randomly partition the $40$ available attributes and into a split of $25/5/10$, which we use to construct the tasks below.

During training, each task is constructed by randomly sampling 2 attributes as Boolean variables and constructing tasks where one class shares the setting of these attributes and the other is the converse. For example, a valid constructed tasks is classifying `not Smiling, Pale Skin` versus `Smiling, not Pale Skin`. During testing, we sample 3 attributes from the test set to form the training task, and sample the 3 corresponding 2-uples to form the test task. After removing combinations that have insufficient examples to form a single tasks, this scheme produces $583/19/53$ tasks for meta-train/val/test respectively. Each sampled image is pre-processed by first obtaining an approximately $168 \times 168$ center crop of each image following by downsampling to $84 \times 84$. This crop is captures regions of the image necessary to classify the non-facial attributes (e.g. Wearing Necklace).

Meta-training attributes:
`Oval Face, Attractive, Mustache, Male, Pointy Nose, Bushy Eyebrows, Blond Hair, Rosy Cheeks, Receding Hairline, Eyeglasses, Goatee, Brown Hair, Narrow Eyes, Chubby, Big Lips, Wavy Hair, Bags Under Eyes, Arched Eyebrows, Wearing Earrings, High Cheekbones, Black Hair, Bangs, Wearing Lipstick, Sideburns, Bald`

Meta-validation attributes:
`Wearing Necklace, Smiling, Pale Skin, Wearing Necktie, Big Nose`

Meta-testing attributes:
`Straight Hair, 5 o'Clock Shadow, Wearing Hat, Gray Hair, Heavy Makeup, Young, Blurry, Double Chin, Mouth Slightly Open, No Beard.`

## B    Experimental Details

In the illustrative experiments, we use a fully connected network with 3 ReLU layers of size 100. Following Finn et al. [10], we additionally use a bias transformation variable, concatenated to the input, with size 20. Both methods use 5 inner gradient steps on $\mathcal{D}^{\text{tr}}$ with step size $\alpha = 0.001$ for regression and $\alpha = 0.01$ for classification. The inference network and prior for PLATIPUS both use one gradient step. For PLATIPUS, we weight the KL term in the objective by $1.5$ for 1D regression and $0.01$ for 2D classification.

For CelebA, we adapt the base convolutional architecture described in Finn et al. [9] which we refer the readers to for more detail. Our approximate posterior and prior have dimensionality matching the underlying model. We tune our approach over the inner learning rate $\alpha$, a weight on the $D_{\text{KL}}$, the scale of the initialization of $\boldsymbol{\sigma}_{\theta}^2, \mathbf{v}_q \in \{0.5, 0.1, 0.15\}$, $\boldsymbol{\gamma}_p, \boldsymbol{\gamma}_q \in \{0.05, 0.1\}$, and a weight on the KL objective $\in \{0.05, 0.1, 0.15\}$ which we anneal towards during training. All models are trained for a maximum of 60,000 iterations.

At meta-test time, we evaluate our approach by taking 15 samples from the prior before determining the assignments. The assignments are made based on the likelihood of the testing examples. We average our results over 100 test tasks. In order to compute the marginal log-likelihood, we average over 100 samples from the prior.

## C    MiniImagenet Comparison

We provide an additional comparison on the MiniImagenet dataset. Since this benchmark does not contain a large amount of ambiguity, we do not aim to show state-of-the-art performance. Instead, our goal with this experiment is to compare our approach on to MAML and prior methods that build upon MAML on this standard benchmark. Since our goal is to compare algorithms, rather than achieving

| MiniImagenet | 5-way, 1-shot Accuracy |
|---|---|
| MAML [8] | $48.70 \pm 1.84\%$ |
| LLAMA [15] | $49.40 \pm 1.83\%$ |
| Reptile [30] | $\mathbf{49.97 \pm 0.32}\%$ |
| PLATIPUS (ours) | $\mathbf{50.13 \pm 1.86}\%$ |
| Meta-SGD [26] | $\mathbf{50.71 \pm 1.87}\%$ |
| matching nets [40] | $43.56 \pm 0.84\%$ |
| meta-learner LSTM [31] | $43.44 \pm 0.77\%$ |
| SNAIL [28]* | $45.10 \pm 0.00\%$ |
| prototypical networks [37] | $46.61 \pm 0.78\%$ |
| mAP-DLM [37] | $49.82 \pm 0.78\%$ |
| GNN [13] | $\mathbf{50.33 \pm 0.36}\%$ |
| Relation Net [38] | $\mathbf{50.44 \pm 0.82}\%$ |

Table 2: Comparison between our approach and prior MAML-based methods (top), and other prior few-shot learning techniques on the 5-way, 1-shot MiniImagenet benchmark. Our approach gives a small boost over MAML, and is comparable to other approaches. We bold the approaches that are above the highest confidence interval lower-bound. *Accuracy using comparable network architecture.

maximal performance, we decouple the effect of the meta-learning algorithm and the architecture used by using the standard 4-block convolutional architecture used by Vinyals et al. [40], Ravi and Larochelle [31], Finn et al. [9] and others. We note that better performance can likely be achieved by tuning the architecture. The results, in Table 2 indicate that our method slightly outperforms MAML and achieves comparable performance to a number of other prior methods.