[Reviews · NeurIPS 2018]

Reviewer 1



Response to the rebuttal: Thanks for the helpful clarifications and new experiments that directly address some of my original comments. I have updated my score from a 6 to a 7. --- This paper proposes a way of modifying MAML to be probabilistic by injecting noise into gradient descent. The method is empirically validated on some synthetic domains and in ambiguous image classification with celebA. I like that the proposed method is also model-agnostic way of adding stochasticity by adding the noise in the parameter space instead of a model-specific way of adding stochasticity. This lets the technique be applied to classification and regression tasks without any modification. I think it would be interesting to further quantitatively study the distributions that this method induces for specific tasks, and to compare it to model-specific approaches. Because this method is straightforward to apply in all of the domains that MAML captures, I am surprised that there are not more quantitative comparisons, especially on the standard few-shot classification tasks such as Omniglot and MiniImagenet. Did you try ProMAML in these domains?

Reviewer 2



This paper proposes a probabilistic meta-learning method that extends MAML to probabilistic version with prior parameters defined on the model. The motivation is to use probabilistic method to capture the uncertainty that is disregarded in deterministic gradient descent methods. Variational inference is adopted to solve the problem and specifically a variational lower bound is used to approximate the likelihood to make the computation tractable. Some regression and classification problems are studied in the experiments. I would agree that the proposed probabilistic MAML method is able to deal with more uncertainty for meta-training compared with MAML, while I have some concerns below. The experimental section considers two toy problems and an image classification problem while all of them are relatively simple cases with regard to their problem scale. In the image classification problem, the coverage metric that “receive at least one sample for each set” is somewhat loose that it is not a strong criterion to access the performance of MAML and ProMAML. The ProMAML method is studied only for some simple supervised learning problems, while in the MAML paper both supersized and RL settings are studied and evaluated. I think it could be much more interesting to apply ProMAML to RL problems many of which may be considered with much more ambiguity and uncertainty. Minor: The subfigures in figure 1 should be subscripted with (a), (b) and (c) which are referenced in the main text. In figure 1, should “p(\phi|x_i^train, y_i^train, \theta)” on the top of the middle subfigure be “p(\phi_i|x_i^train, y_i^train, \theta)”? There is a “?” in the references in line 136. Line 158: “a inference network” -> “an inference network” Line 178: Eq. (1) does not have the second line.

Reviewer 3



This paper presents an extension to the popular metalearning algorithm MAML, in which it is re-cast as inference in a graphical model. This framing allows samples to be drawn from a model posterior, enabling reasoning about uncertainty and capturing multiple modes of ambiguous data, while MAML can only make a single point estimate of model parameters at test time. This is shown in several experiments to better capture the characteristic of ambiguous, noisy data than MAML. Strengths: + The paper makes a strong point that few shot learning is often too ambiguous to confine to a single-model metalearning paradigm. Especially with the high level of recent interest in topics such as safe learning, risk-aware learning, and active learning, this is a relevant area of work. + The graphical model formulation logically follows from the stated goals of the paper, and the inference methods are built on a solid foundation of well-understood recent advances in variational inference and stochastic optimization and appear to be sound. + The paper is well-written on the whole and easy to follow. + The experiments do a nice job at showing intuitively various ways that ProMAML can represent uncertainty -- that it reduces uncertainty as more data is seen, that is can accurately predict uncertainty in a single-shot learning case, and that it captures multiple modes in ambiguous situations. Weaknesses: - The primary weakness of the paper is that the benefits of being able to represent and reason about model uncertainty are never shown or explored experimentally. The experiments do a nice job of showing that uncertainty *can* be represented with some degree of fidelity, but it is never used directly for anything. The paper hints at applications in active learning, risk-aware learning, etc., but never show any of these experimentally. Not only would a concrete example make the paper more compelling, but having a more objective downstream success metric would make it much easier to interpret the results in the paper -- for example, in the final experiment, is coverage of 2.62 and accuracy of ~68% good enough? It depends on the downstream use of the results of ProMAML to answer that question. The authors' response addressed several of my concerns with the new experiments that were shown, particularly my concerns about a "downstream task" that utilizes the uncertainty measure.

Reviewer 4



This paper builds on the probabilistic interpretation of MAML (e.g. Grant et al.) The proposed inference procedure allows for meta-learning for few-shot learning problems in which the small amount of data makes the given task ambiguous. The paper’s writing is clear. The theory is relevant to extending MAML to a distribution over parameters. Experiments highlight the advantages (dealing with ambiguity in tasks) of the proposed method.